# Insights into Gold Nanoparticles Possibilities for Diagnosis and Treatment of the Head and Neck Upper Aerodigestive Tract Cancers

**DOI:** 10.3390/cancers15072080

**Published:** 2023-03-30

**Authors:** Lídia M. Andrade, Guilherme M. J. Costa

**Affiliations:** 1Laboratory of Cell Biology, Department of Morphology, Universidade Federal de Minas Gerais (UFMG), Belo Horizonte 31270-901, Brazil; 2Nanobiomedical Research Group, Department of Physics, Universidade Federal de Minas Gerais (UFMG), Belo Horizonte 31270-901, Brazil

**Keywords:** HNSCC, nanomedicine, gold nanoparticles, theranostic, image-guided surgery, cancer

## Abstract

**Simple Summary:**

The application of nanotechnology in medicine has caught the scientific community’s attention. Nevertheless, staying updated about its countless possibilities has become a challenge. Regarding novelties in the diagnosis and treatment of head and neck cancer, gold nanoparticles have the opportunity to play an important role in nanomedicine. The literature’s findings have provided insights into their applications for research and procedures in oncology. Thus, gold nanoparticles can improve patients’ quality of life and may change paradigms in this field. Therefore, a state-of-the-art review may help clinicians, students, researchers, and readers to keep up with the advances in managing upper aerodigestive tract tumors through gold nanoparticle properties.

**Abstract:**

Background: Head and neck squamous cell carcinoma (HNSCC) is the sixth most common cancer affecting people and accounts for more than 300,000 deaths worldwide. Improvements in treatment modalities, including immunotherapy, have demonstrated promising prognoses for eligible patients. Nevertheless, the five-year overall survival rate has not increased significantly, and the tumor recurrence ratio remains at 50% or higher, except for patients with HPV-positive HNSCC. Over the last decades, nanotechnology has provided promising tools, especially for biomedical applications, due to some remarkable physicochemical properties of numerous nanomaterials, particularly gold nanoparticles. This review addresses the features and some applications of gold nanoparticles reported in the literature over the last five years regarding the diagnosis and treatment of head and neck cancer, highlighting the exciting possibilities of this nanomaterial in oncology. Methods: The scientific papers selected for this review were obtained from the PubMed Advanced, Web of Science, Scopus, ClinicalTrials.gov, and Google Scholar platforms. Conclusions: Results from papers applying gold nanoparticles have suggested that their application is a feasible approach to diagnostics, prognostics, and the treatment of HNC. Moreover, phase I clinical trials suggest that gold nanoparticles are safe and can potentially become theranostic agents for humans.

## 1. Introduction

Head and neck cancers (HNC) are a complex category of tumors belonging to the upper aerodigestive tract, affecting the oral cavity, oropharynx, larynx, upper esophagus, sinuses, salivary glands, and lips, developed mainly due to tobacco and alcohol consumption or HPV infection. Conventionally, the treatment consists of surgery followed by radiotherapy or chemotherapy, alone or in combination, and, where appropriate, epidermal growth factor receptor (EGFR) inhibition [1]. Despite these treatments, the enhancement of the 5-year overall survival rate remains a challenge, and recurrence is often observed, making it clear that choices based on a more accurate patient prognosis involving expertise, individualized schedules considering toxicity, morbidity, and functional preservation of the organs must be pursued [2].

Head and neck squamous cell carcinomas (HNSCC) are the most often histologically diagnosed tumor type, and molecular targets have been investigated to improve HNSCC treatment [3]. Likewise, several molecules have become immunotherapeutic agents in ongoing clinical trials or have already been approved as treatments [4]. Due to its overexpression or upregulated activity in HNSCC, the EGFR is the most investigated molecular target, and some strategies to decrease its overexpression have demonstrated reliability in vitro, such as nuclear Ca^2+^ depletion [5] or the blockage of the EGFR downstream proliferation axis by using the monoclonal antibody Cetuximab, one of the immunotherapies approved for the treatment of recurrent/metastatic HNSCC. Although Cetuximab has demonstrated effectiveness, immune-related adverse effects and/or resistance have already been reported [6,7].

Since Michael Faraday synthesized the first gold nanoparticles (AuNPs) in a colloidal solution in the 19th century, the range of possibilities for AuNP applications has increased to their current use in biomedicine. Their unique properties make AuNPs of particular interest, especially to the pharmaceutical and medical industries. Amongst their applications, biosensing [8,9], vaccines [10,11], drug delivery [12], and DNA sequencing [13,14] are of particular note. Furthermore, AuNPs have been investigated to improve cancer diagnosis, treatment, and patient follow-up. Regarding their applications in diagnostic and cancer treatment modalities, we can also highlight photothermal therapy (PTT), photodynamic therapy (PDT), and immunotherapy [15]. Up to now, promising pre-clinical results have encouraged advanced studies in humans, such as the highly successful prostate cancer treatment using AuNPs combined with PTT [16] and a diagnostic and prognostic clinical trial using gold nanoparticles for salivary gland tumors, which is already in progress. Some of these applications for upper aerodigestive tract cancer treatment and diagnosis by applying AuNPs are discussed in this state-of-the-art review. We aimed to provide the latest information on this exciting field to clinicians, surgeons, researchers, and interested readers as well.

## 2. Methods

The papers and information sources used in this review were selected from the last five years and focus on established head and neck tumor cell lines and xenograft models from different research groups. PubMed/MEDLINE advanced, Web of Science, Scopus, and Google Scholar platforms were searched, and publications that were peer-reviewed, had a minimum journal impact factor of 1, and written in English on the investigation of AuNPs for theranostic applications were considered. Clinical trials applying AuNPs for diagnosis and treatment were searched on Clinical Trials.gov. Core search terms included gold nanoparticles, AuNPs, diagnostic, head and neck tumors, theranostic, cancer treatment, photodynamic therapy, photothermal therapy, immunotherapy, and terms relating to cancer and nanobiomedicine.

## 3. Discussion

AuNPs are metallic nanoparticles often produced through a wet synthesis route. The shape and size of each kind of AuNP depend on specific surfactants such as sodium citrate and cetyltrimethylammonium bromide (CTAB), which promote anisotropic growth. Some basic characterization procedures were based on spectroscopic techniques and electron microscopy imaging. Anatomical head and neck features, AuNP synthesis, and characterizations are described below.

### 3.1. Characteristics of Gold Nanoparticles for Biomedical Applications

The upper aerodigestive tract, especially the head and neck regions, is composed of complex anatomical sites that may impair the localization of tumors and therapeutic interventions. Therefore, an optimized approach should be developed to allow efficient diagnosis and facilitate treatments; this is an issue that has been pursued in the nanomaterials field of research. Figure 1 provides anatomic details of the upper aerodigestive tract sites.

Among inorganic nanomaterials, AuNPs have been largely studied for those proposals. They can be synthesized from an *Au* donor, commonly chloroauric acid (HAuCl_4_)_,_ and engineered for different sizes and shapes according to the proposal. Moreover, each type of AuNP possesses characteristic spectra; for instance, spherical AuNPs have their absorption peak at 520 nm, while rod-like AuNPs, known as GNRs, have absorption peaks varying from 620 nm to 1000 nm, depending on their aspect ratio [15]. Low cytotoxicity, a huge volume/area ratio, which allows multiple surface coatings, and collective electron oscillations, recognized as localized surface plasmon resonance (LSPR), are the remarkable characteristics of these nanoparticles. AuNPs can absorb and scatter light within the visible light range, which is one of their most remarkable optical properties for biomedicine and enables them to interact with molecules in their vicinity. At the nanometer scale, their sensitivity is enhanced exponentially, and any minimal alteration can be detected and measured via dynamic light scattering, a property known as plasmonic shifts. Through a conventional plate spectrophotometry reader coupled with ultraviolet–visible light (UV-Vis), it is possible to perform diagnostics using AuNPs LSPR shifts [8,17], as well as via other spectroscopic techniques and tools; most of these spectrophotometers are already available for purchase. Another feature worth mentioning is the colorimetric changes, observable to the naked eye, that generate spectroscopic signatures that work as fingerprints, allowing their use for diagnosis and therapy. Additionally, AuNPs’ surface modifications are performed to enhance their bioavailability, as well as to protect the nanoparticle from the environment and efficiently target a specific tissue or cell type.

The stable biofunctionalization of AuNP surfaces with antibodies and drugs requires very low concentrations. Nanocarrier molecule pharmacokinetics pass through biological barriers faster than conventional medicines, decreasing toxicity, efficiently targeting tumor sites, and increasing the time of nanoparticle retention close to the tumors in a process known as enhanced permeability and retention (EPR) [18,19]. This is advantageous because it permits the nanoparticles to reach and access the tumor microenvironment (TME), contributing to the achievement of highly effective treatments.

There are some natural mechanisms involved in AuNPs’ cellular targeting and internalization through the cell membrane. The most common cellular internalization encompasses endocytic pathways such as clathrin- and caveolin-dependent/independent endocytosis, micropinocytosis, and phagocytosis [20]. In addition, transmembrane receptors work to deliver and transport functionalized AuNPs, increasing their cellular uptake [21]. Notwithstanding, several challenges exist for successful AuNP cellular uptake related to their physicochemical properties, surface chemistry, stiffness, size, and shape. Additionally, human body barriers present remarkable biological features that may impair AuNPs’ internalization and trafficking. For instance, the desirable outcomes of the various ionic interactions with the cell membrane can be modified by the size and surface charge interactions of sphere-shaped AuNPs. Homogeneity and carefully controlled aspect ratio synthesis, which are difficult, are required for rod-shaped AuNPs to obtain a good uptake due to reduced surface energy needs in cell membrane wrapping [19,20,22]. Some of the most relevant physical characteristics of AuNPs are displayed in Figure 2.

### 3.2. Gold Nanoparticles for Diagnostic Proposals

One of the biggest challenges for surgeons is to have a clear picture of where the tumor ends and the healthy tissue begins [23]. Therefore, functionalized AuNPs have been engineered to target tumor tissues, draw the borders where normal tissues remain, provide targeted molecular imaging, enable intraoperative identification of surgical resection margins, and monitor oral cancer prognosis after treatment [24]. Due to their abilities to simultaneously be diagnostic and treatment agents, the term theranostic was coined. For example, nanoparticle-based contrast agents have been indicated as powerful tools for enhancing in vivo imaging. On the other hand, it has been suggested that therapeutic nanoparticles may overcome several limitations of conventional drug delivery systems [25].

Taking advantage of their countless spectrometry applications, AuNPs have been pointed out as a helpful approach for spectroscopic cancer diagnostics. One approach, for instance, applies surface-enhanced Raman spectroscopy (SERS), which allows the identification of molecular vibration signatures, for which AuNPs play a pivotal role. Blood sera samples from 135 patients with oral squamous cell carcinoma and mucoepidermoid carcinoma were exposed to AuNPs and compared to the sera of healthy donors. After measuring the samples via SERS, the authors found a sensitivity of 80% and a specificity of 84% in discriminating nucleic acids and some proteins from patients’ sera. The authors suggested that this technique presents sensitivity enough for an early diagnosis of HNSCC [26] and can also be applied to cancer staging systems (tumor, lymph node, and metastasis—TNM) for oral squamous cell carcinoma (OSCC) classification based on SERS features obtained from the patient’s blood sera [27].

As a novel class of contrast agents for fluorescence-based detection of HNC, AuNPs can work as a molecular contrast agent that alters the fluorescent spectra of tumor cells. AuNPs can scatter light several times stronger than fluorescent dye emission [28], enabling fluorescence lifetime imaging microscopy (FLIM) to measure and quantify the fluorescence from biological samples. Balb/C mouse organs were exposed to AuNPs coated with the fluorophore Oregon Green 488-X. The data showed detection and measurements of fluorescence independent of pH or enzymes present in different tissues [29]. In addition, Reis et al. have demonstrated that AuNPs enable enhanced fluorescence signal detection through flow cytometry in squamous cell carcinoma (SCC) A431 cells compared with cellular labeling standard protocols, suggesting that AuNPs can be used for fluorescence-based diagnostic and cellular staining [30].

In another study, gold nanoclusters were developed for image-guided surgery using a near-infrared (NIR) optical live imaging system to localize tumor residues, allowing their identification through the fluorescence of small remaining tumor lesions after surgical removal. It demonstrated a tumor/skin imaging ratio of >1.5 and an increase in survival time of 35% compared to conventional optical-guided surgery methods [31].

A xenograft nude mouse nasopharyngeal HNC model was used for molecular computed tomography (CT) imaging using AuNPs coated with folic acid and cysteamine. The data demonstrated that this molecular CT imaging approach was sensitive enough to detect even small tumor sizes through enhanced X-ray signals, and the authors also suggested this application for use in micrometastasis screening [32].

Epidermal growth factor receptor expression is one of the most important biomarkers for HNSCC diagnosis, customarily obtained via immunohistochemistry. Liu et al. developed a terahertz metamaterial biosensor device constructed with AuNPs and EGFR antibodies, which demonstrated a larger frequency shift range and a higher sensitivity detection for EGFR overexpression, even under low sample concentrations [29].

A specific biosensing system for detecting human papillomavirus (HPV) using AuNPs and the DNA complementary sequence of the E6 HPV-16 oncogene was developed. As a proof of concept, this approach was employed using samples obtained from eight patients with histologically confirmed HNSCC. The data were validated via real-time quantitative polymerase chain reaction (RT-qPCR), which demonstrated the sensitivity of this nanosensor and suggested its use as a diagnostic tool for screening positive HPV-related HNSCC cases [33]. There are many other possibilities for HNC diagnostic applications of AuNPs, such as photoacoustic imaging, colorimetric-based assays, and bio-barcoding, which have already been detailed in other studies [28,34,35]. The characteristics and applications of different gold nanoparticles for diagnoses and treatments are described in Table 1.

### 3.3. Gold Nanoparticles for Tumor Treatment

Radiation oncology is a conventional modality for the treatment of HNC. Despite its outstanding curative outcomes, radioresistance has been reported, and sequelae such as mucositis and xerostomia often occur. AuNPs have been considered potential sensitizers in radiotherapy to improve tumor radiation effects and minimize undesired side effects. Due to their high atomic number, AuNPs produce strong photoelectric absorption and secondary electron emission caused by gamma- or X-ray irradiation. The main mechanisms identified as being involved in the biological response of cells to AuNPs radiosensitization are the production of oxidative stress, DNA damage induction, cell cycle arrests, and potential interference with the bystander effects [49,50]. Additionally, AuNPs can amplify photons and particles that cause ionizing radiation effects. Kim et al. demonstrated the effectiveness of AuNPs in amplifying neutron radiation cytotoxicity in tumor cells, probably due to a high linear energy transfer [51].

To achieve any enhancement in radiotherapy (RT), AuNPs must be delivered to and internalized by cancer cells. They enter cells through receptor-mediated endocytosis (RME) [39]; thus, combining AuNPs with radiotherapy and chemotherapy is promising for a complete treatment modality [40], promoting tumor cell death mainly due to apoptosis [36].

Some strategies to facilitate AuNP internalization and therapeutic improvements encompass functionalization with chemotherapy agents. A nanocomplex containing AuNPs functionalized with Cisplatin was proposed to work as a theranostic platform in combination with RT. This approach demonstrated significant in vivo tumor inhibition after irradiation and its feasibility as a CT contrast agent [37]. Another work demonstrated enhanced cytotoxicity with decreased cell proliferation and increased apoptosis by applying AuNPs + X-rays and AuNPs + epidermal growth factor inhibitors combination in the HNSCC cell line model [38].

Interestingly, in a study, AuNPs were coated with Crizotinib, an anaplastic lymphoma kinase (ALK) inhibitor, for targeting and radiosensitization of adenoid cystic carcinomas (ACC). A sample from a patient with ACC of the parotid gland and its tumor molecular profiling identified an AKL mutation. Next, this sample was subcutaneously injected into nude mice and treated with AuNPs coated with Crizotinib, followed by irradiation. The data showed gold concentration in the tumor mass, a reduction in tumor volume, DNA damage, and apoptosis in the group treated with this nanocomplex [52].

AuNPs’ enhanced radiosensitization effect is achieved better at kiloelectron volt (KeV) energies than at the clinical megavoltage (MV) used in RT. Thus, Piccolo et al. developed a diamond target beam (DTB) system to enhance the quantity of low-energy photons to achieve AuNPs in vitro by applying head and neck tumor cell lines and in vivo using a zebrafish xenotransplantation model. The authors observed decreased HNC cellular viability in vitro and enhanced tumor cell killing in zebrafish xenografts compared to standard RT, mainly due to DNA double-strand breaks and increased reactive oxygen species (ROS) production, mostly superoxide anions, hydroxyl radicals, and hydrogen peroxides [53]. These pre-clinical data suggest that AuNPs have a great potential to improve radiation oncology, with the possibility of conversion into clinical benefits for patients in the following years.

Tumor cells are sensitive to temperature differences, making hyperthermia a cancer treatment modality. When the light in the NIR region reaches inorganic nanoparticles, electromagnetic energy is converted into heat, known as photothermal therapy (PTT); this is one of the physical properties of AuNPs, allowing for their use in thermal cancer treatment. For PTT applications, some prerequisites are required, such as good dispersibility in aqueous solutions, photostability, a sufficient tissue penetration depth, and a response to light in the NIR range of 650–950 nm [54].

Zhang et al. synthesized gold nanorods (GNR) coated with EGFRmAb and applied a NIR laser at 808 nm for 6 min to previously treated larynx cancer cells, obtaining an average local temperature of 50 °C. They observed cell death by apoptosis [41]. Likewise, GNR coated with Cetuximab and the human anti-EGFR antibody EGFR-hIgG1 were applied to the HNC cell line CAL-27. After 24 h of treatment, cells were irradiated using a continuous wave diode laser at 1064 nm with a power of 2 Wcm^−2^ for 2 min. The increased local temperature associated with significant tumor cell death was observed, and the authors suggested targeting EGFR combined with GNR as a suitable approach for PTT [42].

In a similar interesting approach, selectivity macrophages have been proposed as a cellular vector for delivering nanoparticles. Gold nanoshell- and pegylated-gold nanoshell-loaded macrophages were co-cultured with tumor cells and exposed to 810 nm or 960 nm NIR irradiation, reaching a local temperature of ~45 °C and enhancing cytotoxic effects [43].

By using HNSCC spheroids, Mapanao et al. demonstrated that gold-nanoarchitectures could promote hyperthermia and disassemble into the building blocks, releasing available prodrug molecules simultaneously [44] and working as a combined photothermal-chemotherapy. Similarly, hybrid spheroids were generated with gold nanoshells using a macrophage-mediated delivery system and the FaDu cell line, a human hypopharynx HNSCC. These spheroids were irradiated with a NIR laser, and the PTT-treated hybrid model induced increased apoptosis and dead cells [45].

Furthermore, inorganic multifunctional nanoparticles can offer a new approach to treating HNSCC by combining plasmonic resonance with targeting antibodies. Then, AuNPs functionalized with immunotherapeutics can deliver local hyperpyrexia (caused by NIR) to treat both primary and metastatic malignancies, resulting in a novel therapeutic approach known as photoimmunotherapy (PIT) [55].

Another modality of cancer treatment is photodynamic therapy (PDT). This promotes increased levels of reactive oxygen species (ROS) after NIR irradiation, mainly producing singlet oxygen, which is toxic for nucleic acids. AuNPs coated with photosensitizers, such as porphyrin [56], can be applied to treat superficial epidermoid carcinomas and have been suggested to treat mucosal and tongue tumors. Duman et al. developed a nanocomposite based on GNR encapsulated with polyacrylic acid (PAA), which was electrostatically bound to the cationic porphyrin TMPyP. The photosensitizer effect of this approach was assessed in 2D and 3D HNSCC models. These culture systems were illuminated with peak emission at 420 nm wavelength and 7 mWcm^−1^ for up to 10 min. Intense phototoxicity was observed, leading to decreased tumor cell proliferation in both models [46]. In another work, HNSCC cell lines were exposed to AuNPs coated with 5-aminolevulinic acid and irradiated with a red light-emitting diode (LED) source at 621 nm for PDT. This approach led tumor cells to apoptosis and suppressed mRNA and protein expression levels of β-catenin, c-myc, and cyclin D1, respectively [47]. Accordingly, AuNPs’ physical properties enable simultaneous therapeutic purposes by combining PDT and PTT. Thus, Gong et al. developed a new nanoplatform from a thermo-responsive polymer-encapsulated GNR incorporating indocyanine green (ICG) to enhance the photodynamic/photothermal therapy effects [57].

Several immunotherapy agents have been approved for treating head and neck cancer, including antibodies targeting EGFR and immune checkpoint inhibitors for managing recurrent or metastatic tumors. EGFR inhibition schemes have been studied to discover and validate reliable strategies to hamper signaling pathways in heterogenous HNC and to overcome anti-EGFR resistance [6]. For instance, Cetuximab, Nivolumab, and Pembrolizumab are currently approved for HNC treatment, and, despite their limitations in treating these kinds of tumors, the efficacy of these antibodies could be enhanced through conjugation to AuNPs [7]. Our research group has demonstrated the effects of AuNPs coated with a low concentration of Cetuximab, an EGFR inhibitor, showing improved apoptosis in A431 cells compared with Cetuximab alone [48].

Programmed cell death protein 1 (PD-1) [2] is a member of the CD28 receptor family, which is expressed on activated T and B cells, monocytes, and a subset of thymocytes that work as an inhibitor of T cell responses [58]. Anti-PD-1 immunotherapies have been developed, and some clinical trials for this target are in progress. The main drawback to an effective immunological response against cancer cells is the TME, where immune evasion processes often occur [59].

As an attempt to overcome these processes, AuNPs have been thought to act as immune regulators and an effective immune-delivering drug for cancer, making them appropriate candidates for enhancing the efficiency and safety of cancer immunotherapy [60] and modulating TME [61]. In addition, AuNPs can work as artificial antigen-presenting cells (APCs) and adjuvants by binding with dendritic cells (DCs) and activating the production of cytokines [62]. The interactions between AuNPs and anti-PD-1 antibodies have been investigated in other tumor models, and the results have demonstrated increased apoptosis and inhibition of tumor cell-mediated angiogenesis [19,61,62].

Numerous drug delivery systems, based on nanotechnology, for the treatment of HNC have been developed to improve the potential therapeutical effects of drugs through a targeted strategy and better bioavailability at cancer sites [63]. The applications of AuNPs for drug delivery have been extensively discussed in the literature, and complete reviews can be easily found elsewhere. Figure 3 summarizes the in vitro effects of AuNPs and the therapeutic strategies described in this review.

In this state-of-the-art review, 66.6% of the AuNPs studied in research papers applied were sphere-shaped, followed by 12.5% that were rod-like shaped. Regarding biomedical proposals, 58.4% were studied for therapy, mostly involving RT and PTT (35.7%), followed by PDT and IT (14.3%), as demonstrated in Figure 4.

### 3.4. Implication for Practice

Head and neck cancer causes pain, orofacial dysfunctions, and death, pushing science and medical communities forward to improve early diagnosis and treatment tools, focusing on maintaining the patient’s quality of life. In this way, theranostics has arisen as a new field for the next generation of medicine and was revealed thanks to nanotechnology.

AuNPs have demonstrated many possibilities for various purposes from drug delivery to in loco physicochemical treatments. In this context, nanomedicine research has a direct impact on HNC. Due to their unique optical properties, AuNPs represent an important nanomaterial for biomedical procedures. Even when combined with other nanomaterials, AuNPs may help increase cellular biocompatibility [64]. Some level of cytotoxicity appears when AuNPs are used in biological assays. One common concern addresses tissue AuNPs aggregation that could lead to local necrosis. Therefore, suitable chemical surface modifications with PEG have been applied to improve AuNPs’ bioavailability [65]. Additionally, AuNP toxicity is strongly related to by-products generated during the synthesis and functionalization processes, type of surfactants and their concentration, route of administration, size, charge, and shape [66]. For instance, it has already been demonstrated that the smallest nanoparticles are the most cytotoxic [67,68].

Another critical point is the lack of standard AuNP synthesis and functionalization protocols. Likewise, a well-established universal safety regulation based on reliable and robust multicentric research should be taken into account, and the international community must make an effort to adopt specific protocols for the characterization of these products [69]. In this way, the European approach to the regulatory testing of nanomaterials has tried to promote a comprehensive consensus framework for the safety assessment of nanomaterials (NaNoREG) [70].

Especially in oral oncology, standard-of-care treatments such as chemotherapy and radiotherapy nanomedicine have arisen as new perspectives to improve outcomes; fields that can benefit from incorporating AuNPs, for example, through Cisplatin-coating. Moreover, imaging-guided surgery may help surgeons avoid residual tumor margins through enhanced fluorescence surrounding tissue borders.

There is a trend in the use of nanotechnology to enhance tumor immune effects. The nanoparticle-based delivery system can activate dendritic cells or T cells for immunotherapy [59,61,62], and the application of the immunostimulatory nanoparticles as a smart carrier for the effective delivery of cancer antigens and adjuvants can improve tumor immunology responses [71]. Even though 37.5% of the data found have shown unspecific targets, EGFR inhibition molecules are the most conjugated for immunotherapy proposals to AuNP surfaces [72]. They were the most investigated target in the research papers analyzed in this review, corresponding to 25%, and are strong candidates to become a nano-immunotherapy for HNSCC treatment (Figure 5). Nevertheless, the possibility of tailoring immune responses using precisely engineered chiral inorganic nanostructures can lead to a better understanding of their role in biological systems [11].

Three early phase I clinical trials applying AuNPs focused on therapy and diagnosis for tumors from the upper aerodigestive tract region. Safety and pharmacokinetics were assessed in these studies, which encompassed brain, gastric, and salivary cancers. Kumthekar et al. developed an approach based on AuNP cores covalently conjugated with radially oriented and densely packed siRNA oligonucleotides (NU-0129). Patients with recurrent glioblastoma multiforme were treated with intravenous administration, and the authors found the presence of Au in all tumor specimens analyzed. Likewise, the downregulation of Bcl2L12 and caspase-3 inactivation were observed [73]. This work was part of an interventional study (NCT03020017) that aimed to evaluate the safety of NU-0129, and its data showed no significant treatment-related toxicities [74].

AuNPs coated with the protein-coding gene molecule CD24 were part of the trial NCT04907422 [75]. In this study, a novel diagnostic and prognostic approach was developed for the early detection of cancer stem cells in salivary gland tumors. The diagnostic was based on CD24-gold nanocomposite expression using RT-qPCR. This research has been completed, and its data analysis is in progress.

The study of the exhaled breath and salivary metabolites of patients with malignant or benign gastric lesions, NCT01420588 [76], used a designed nanosensor based on AuNPs and samples collected from saliva and alveolar exhaled breath. The investigators analyzed the chemical composition of the samples using gas chromatography coupled with mass spectrometry (GC-MS), and the data are still under evaluation. These are starting trials, but their preliminary results have shed light on the possibilities of AuNPs in oncology and how they work inside human bodies. Detailed information regarding these clinical trials can be found in Table 2.

Animal models and clinical trials can answer critical questions regarding the safety and tumor response of bare or surface-coated AuNPs. Despite the small amount of recent in vivo data, similar results have been demonstrated, mainly promoting DNA damage, apoptosis, and decreased tumor cell proliferation. Figure 6 summarizes the most common cellular findings after AuNPs treatments.

Considering the usual treatment modalities for HNC, RT seems to benefit from greater advantages in AuNP applications, considering the number of studies reported in the literature. On the other hand, new approaches such as immunotherapy have gained traction through several efforts to overcome immune-related toxicities and enhance therapeutic effects through their functionalization with AuNPs. The knowledge of the main mechanisms involved in cancer cell death when therapeutic candidates are evaluated is crucial. Therefore, using AuNPs, cellular death due to apoptosis appeared in 70% of the research papers analyzed, followed by 23% of DNA double-strand breaks, as shown in Figure 7.

Another bottleneck is translating AuNPs from the laboratory scale to industry, and, up until now, there are only a few companies producing AuNPs at the industrial level. Drawbacks such as batch-to-batch reproducibility are challenging [77]. In order to address these issues, a great deal must be said about the lack of standard operating procedures for the synthesis, functionalization, and characterization of AuNPs for biomedical applications, as well as the price of production, stability, and availability of raw materials. Colloidal stability can be achieved by observing water quality, storage temperature, and solution sterility. Regarding shelf-time, the long-term stability of sphere-shaped AuNPs coated with the monoclonal antibody Cetuximab was already observed for up to 24 months [48]. Despite the challenges involved in the development of nanomedicines at industry levels, this field presents viable opportunities for improving the treatment and diagnosis of HNC [78].

Over the last five years, the number of relevant published findings involving AuNPs and upper aerodigestive tract tumors has not increased as it should when taking into account the advances in the technology available. Many data came from in vitro research that presents some limitations, for example, in terms of AuNPs biodistribution assessment. On the other hand, in vivo data still need different animal models focusing on long-term effects to support new clinical trials. Perhaps the COVID-19 pandemic could have impacted researchers’ abilities to continue working at their laboratories, which may explain the reduction in publications on this topic. Nevertheless, it is still possible to assume that this field has a promising outcome. Although pre-clinical studies have suggested favorable results and early phase I trials are currently ongoing, the translation to the clinic will demand more bio–nano interface research, circumventing possible systemic toxicities [79,80], and more clinical trials showing long-term follow-ups are required. An overview of the main possibilities of AuNPs for HNC theranostics is depicted in Figure 8.

## 4. Conclusions

It is necessary to work to provide safe and effective methods for managing HNC since patients who undergo traditional tumor therapies have high levels of complicated morbidities. Among them, xerostomia, facial nerve paralysis, ageusia, mutism, and facial disfigurement directly impact patients’ quality of life. Another important goal is to achieve higher scores of curability, extending the time of free-survival disease, because this scenario has not changed over the last decades, mainly due to late diagnosis, which is especially a common problem in low-income countries. An approach that could simultaneously provide diagnosis, treatment, and follow-ups using the same tools has been of interest and sought after.

The increase of affordable nanomaterials for biological applications can shorten the time from laboratory research to hospital application, and AuNPs seem to be one of the most exciting nanomaterials for achieving this goal. Their extraordinary physicochemical properties make them eligible for theranostic proposals, working as nanocarriers, radiosensitizers, and fluorescent enhancers. The in vitro and in vivo results obtained from different research groups appear promising, even though some bottlenecks regarding reproducibility from batch to batch and related toxicity still need to be overcome. Another critical point is related to regulatory issues for human application. However, despite their small number, clinical trials are working toward the approval of AuNPs for human therapy. Nevertheless, clinicians and researchers dedicated to HNC management can glimpse the exciting possibilities gold nanoparticles offer in the oncology field and the incorporation of nanomedicine as a trend for the future.

## Figures and Tables

**Figure 1 cancers-15-02080-f001:**
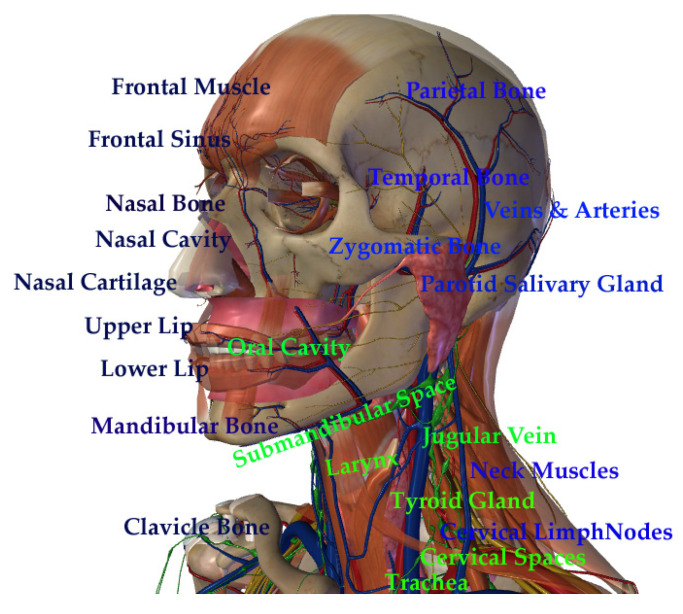
Schematic drawing of the head and neck anatomical sites (adapted from Zygote body 3D Anatomy).

**Figure 2 cancers-15-02080-f002:**
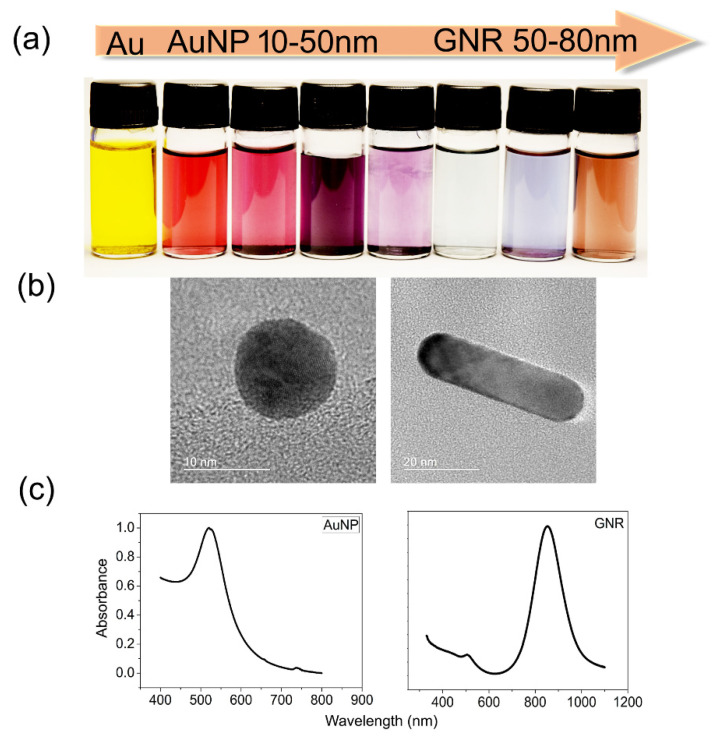
Optical properties of AuNPs. Panel (**a**) shows “colored” nanoparticles due to their interaction with light. Panel (**b**) shows representative transmission electron microscopy images of an AuNP and a GNR with different shapes and sizes. Panel (**c**) shows the absorption spectra of AuNP at 520 nm and GNR at 850 nm, respectively. Au = gold; AuNP = gold nanoparticle; GNR = gold nanorod. Data were produced in our laboratory.

**Figure 3 cancers-15-02080-f003:**
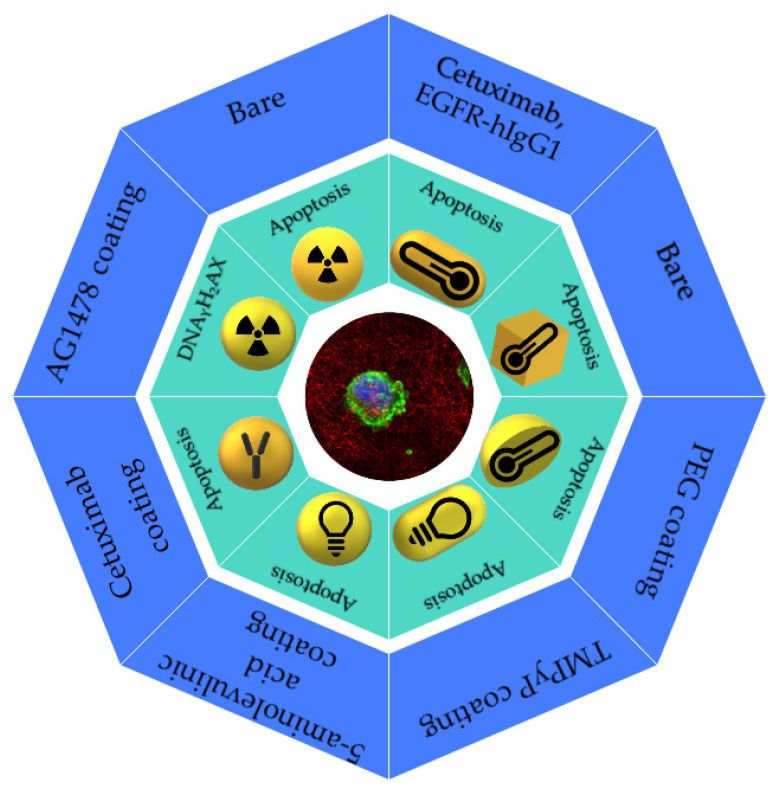
Scheme of several applications of functionalized AuNPs. The shapes (sphere, rhomboid, shell, and rod) and functionalization of the gold nanoparticles utilized in biomedical applications and their effects on cancer cells.

**Figure 4 cancers-15-02080-f004:**
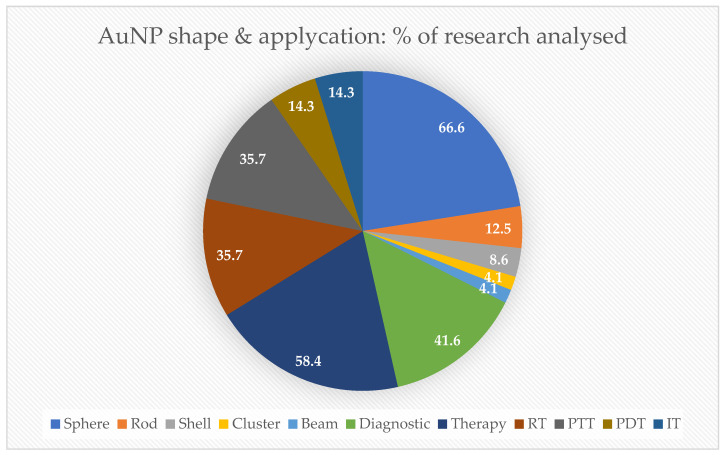
The gold nanoparticles’ shapes and proposals. The percentage of the most common AuNPs shown in research papers used in this state-of-the-art review for biomedical applications. IT = immunotherapy; PDT = photodynamic therapy; PTT = photothermal therapy; RT = radiotherapy.

**Figure 5 cancers-15-02080-f005:**
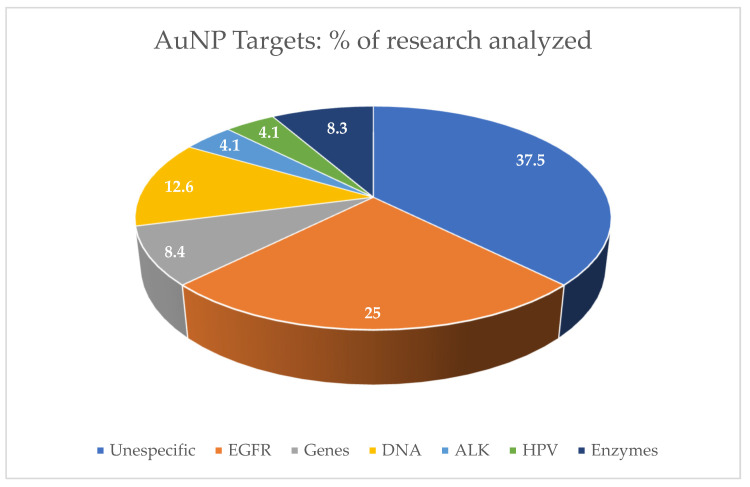
Targets for AuNP research. The percentage of the studied cellular targets using AuNPs for biomedical applications in research papers in this state-of-the-art review. HPV = human papillomavirus; ALK = anaplastic lymphoma kinase; EGFR = epidermal growth factor receptor.

**Figure 6 cancers-15-02080-f006:**
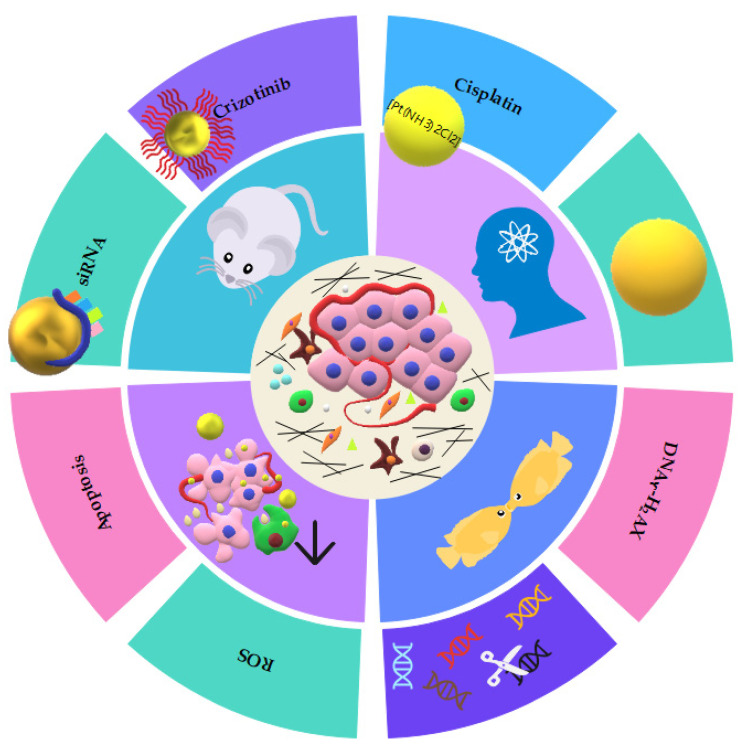
Scheme of in vivo effects of gold nanoparticles. DNA damage, apoptosis, and the elimination of tumor cells are promoted by AuNPs functionalized with chemotherapeutic antibodies or siRNA. siRNA = short interference RNA; ROS = reactive oxygen species.

**Figure 7 cancers-15-02080-f007:**
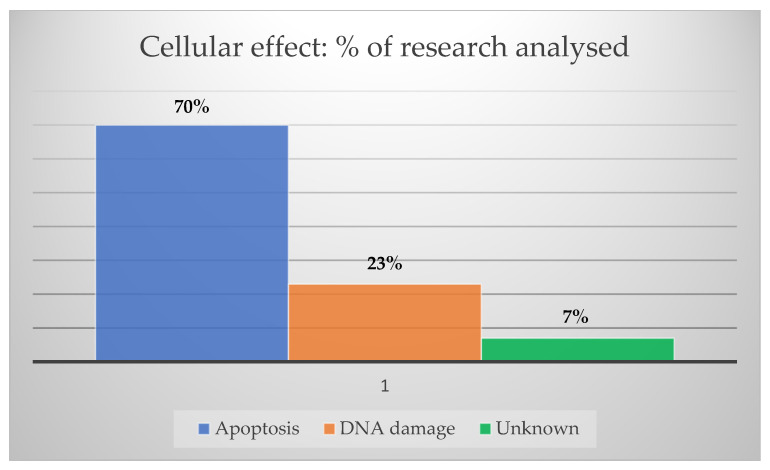
Cellular death mechanisms. The percentage of the most common cancer cellular death mechanisms by AuNPs in research papers used in this state-of-the-art review.

**Figure 8 cancers-15-02080-f008:**
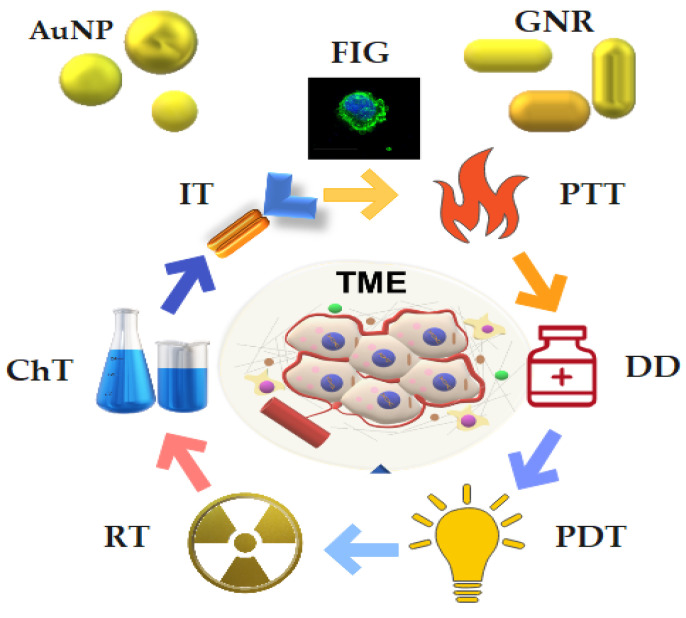
Theranostic applications of gold nanoparticles. AuNPs used for biomedical applications are sphere-shaped (AuNP) and rod-shaped (GNR). TME = tumor microenvironment; RT = radiotherapy; PTT = photothermal therapy; PDT = photodynamic therapy; IT = immunotherapy; ChT = chemotherapy; DD = drug delivery; FIG = fluorescence image-guided therapy.

**Table 1 cancers-15-02080-t001:** AuNPs applied for HNC theranostic proposals.

Nanoparticle Features	Purpose	Target	Study Design	Tumor Death	Reference
Sphere-shaped (~20 nm).	Diagnostic	Nucleic acids and proteins	Human blood samples from patients with mucoepidermoid carcinoma, oral squamous cell carcinoma, and normal individuals.	_	[25]
Bean-shaped (55 nm).	Diagnostic	Nucleic acids and proteins	Human blood samples from patients with mucoepidermoid carcinoma, oral squamous cell carcinoma, and normal individuals.	_	[26]
Sphere-shaped (~20 nm) coated with Oregon Green 488-X.	Diagnostic	Trypsin	In vitro using healthy male Balb/C mice organs for FLIM image acquisition and measurement.	_	[29]
Sphere-shaped (~20 nm) coated with Cetuximab.	Diagnostic	EGFR	In vitro fluorescence enhancement using human squamous cell carcinoma A431 cell line.	_	[30]
Nanoclusters (1.4 nm to 10.5 nm).	Diagnostic	Tumor cells	In vitro using female athymic NMRI nude mice for orthotopic implantation of the HNC cell lines CAL-33 and SQ20B and fluorescence contrast image acquisition.	_	[31]
Sphere-shaped (~15 nm).	Diagnostic	Tumor cells	In vitro using male nude mice for xenograft implantation of nasopharyngeal adenocarcinoma KB cell line.	_	[32]
Sphere-shaped (5 nm, 15 nm, and 25 nm).	Diagnostic	EGFR	Samples of EGFR antibodies were deposited in a bow-tie device for plasmonic resonance signal biosensing at a frequency of tHz.	_	[34]
Sphere-shaped (~10 nm).	Diagnostic	HPV	Samples from patients with HPV+ oropharyngeal cancer were exposed to nanoparticles to detect the E-6 HPV-16 oncogene.	_	[33]
Sphere-shaped (~5 nm).	Therapy—ionizing radiation	Tumor cells	In vitro using human tongue squamous cell carcinoma cell line HSC-3.	Apoptosis	[36]
Sphere-shaped (~20 nm) coated with glucose and Cisplatin.	Therapy—ionizing radiation	DNA	In vitro xenograft nude mice model using A431 cell line.	DNA _ϒ-_ H2AX	[37]
Sphere-shaped (~60 nm) coated with AG1478.	Therapy—ionizing radiation	EGFR	In vitro using human tongue squamous cell carcinoma cell line HSC3.	Apoptosis	[38]
Sphere-shaped (~20 nm) coated with Crizotinib.	Therapy—ionizing radiation	ALK	In vitro xenograft mice model using ACC sample collected from a patient donor.	DNA _ϒ-_ H2AX	[39]
Sphere-shaped (~15 nm).	Therapy—ionizing radiation	Tumor cells	In vitro using FaDu, HSC-3, Detroit-562 cell lines and in vivo using zebrafish xenotransplantation.	DNA _ϒ-_ H2AX	[40]
Rod-shaped (12 nm × 50 nm) coated with EGFR-mAb.	Therapy—hyperthermia	PI3K/AKT/mTOR	In vitro using human larynx squamous cell carcinoma FaDu cell line.	Apoptosis	[41]
Rod-shaped (10 nm × 67 nm) coated with Cetuximab and EGFR-hIgG1.	Therapy—hyperthermia	EGFR	In vitro using human squamous cell carcinoma CAL-27 cell line.	Apoptosis	[42]
Nanoshell (14 nm) and silica core (157 nm).	Therapy—hyperthermia	Tumor cells	In vitro using macrophages and human squamous pharyngeal SNU-1041 cell line.	Unknown	[43]
Rhomboid (~98 nm).	Therapy—hyperthermia	Tumor cells	In vitro using human squamous cell carcinoma SCC-25 (HPV-negative) and UPCI: SCC-154 (HPV-positive) cell lines.	Apoptosis	[44]
Nanoshell (~152 nm) coated with PEG.	Therapy—hyperthermia	Tumor cells	In vitro using human larynx squamous cell carcinoma FaDu cell line.	Apoptosis	[45]
Rod-shaped (~148 nm) Coupled with TMPyP.	Therapy—photodynamic	Tumor cells	In vitro using human squamous cell carcinoma A431 cell line.	Apoptosis	[46]
Sphere-shaped (~18 nm) coated with 5-aminolevulinic acid.	Therapy—photodynamic	Cytoplasm	In vitro using human larynx squamous cell carcinoma FaDu cell line.	Apoptosis	[47]
Sphere-shaped (~20 nm) coated with Cetuximab.	Immunotherapy	EGFR	In vitro using human squamous cell carcinoma A431 cell line.	Apoptosis	[48]

EGFR = epidermal growth factor receptor; mAb = monoclonal antibody; PEG = polyethylene glycol; ALK = anaplastic lymphoma kinase; HNC = head and neck cancer; tHz = terahertz; FLIM = fluorescence lifetime imaging microscopy; ACC = adenoid cystic carcinoma.

**Table 2 cancers-15-02080-t002:** Clinical trials applying AuNPs for diagnosis and treatment of brain, salivary gland, and gastric tumors.

* Study Phase	Title	Study Design	Study Accession NTC Number
**Early Phase 1** 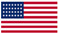	A Phase 0 First-In-Human Study Using NU-0129: A Spherical Nucleic Acid (SNA) Gold Nanoparticle Targeting BCL2L12 in Recurrent Glioblastoma Multiforme or Gliosarcoma Patients	Gold nanoparticles were coated with NU-0129 and arranged in spherical nucleic acids. This platform was infused in patients with glioblastoma multiforme or gliosarcoma. The purpose of this research study was to evaluate the safety of the platform. The nucleic acid component can target a gene called Bcl2L12, which is present in glioblastoma multiforme and is associated with tumor growth. Eight patients were enrolled and received NU-0129 IV over 20–50 min and underwent standard-of-care tumor resection within 8–48 h.	**NCT03020017**Northwestern University[73,74]
** Early Phase 1** 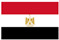	Gold Nanoparticles as Novel Biomarkers for Cancer Stem Cells in Salivary Gland Tumors: A Diagnostic and Prognostic Accuracy Study	Gold nanoparticles were conjugated with the protein-coding gene molecule CD24. This study aimed to introduce a novel diagnostic and prognostic approach in the early detection of cancer stem cells in salivary gland tumors using gold nanoparticles conjugated to CD24 (CD24-gold nanocomposite). Carcinoma ex pleomorphic adenoma of salivary glands and pleomorphic adenoma of salivary glands were tested. This approach was designed for cancer diagnostics through RT-qPCR, and sixty patients were enrolled.	**NCT04907422**October 6 University[75]
** Early Phase 1** 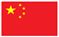	Study of the Exhaled Breath and Salivary Metabolites of Patients with Malignant or Benign Gastric Lesions	The study aimed to test a novel method in oncology based on breath analysis and saliva with a nanosensor array for identifying gastric diseases. Alveolar exhaled breath samples collected from volunteers referred for upper endoscopy or surgery were analyzed using a custom-designed array of chemical nanosensors based on organically functionalized gold nanoparticles. The chemical composition of the breath samples was studied using gas chromatography coupled with mass spectrometry (GC-MS). One thousand patients were enrolled.	**NCT01420588**Anhui Medical University[76]

* Information obtained from https://clinicaltrials.gov/, (accessed on 8 April 2022).

## Data Availability

Not applicable.

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
