# Peer review of "Insights into Gold Nanoparticles Possibilities for Diagnosis and Treatment of the Head and Neck Upper Aerodigestive Tract Cancers"

_cancers, 2023, doi:10.3390/cancers15072080_

Round 1
Reviewer 1 Report
Current review article on the insights of gold NPs as a theranostic tool covers vast, latest, and quality compilation. Authors have presented a well drafted review by collecting information from highly scientific and reputed literature, while their method of selection was greatly unique. Authors have widely focused on the different aspects of the gold NPs in terms of therapeutic approach, applications, current status and several case studies. Overall, the efforts in the current review are enormous. There are only few minor comments and suggestions to improve the final quality of the review after which it can be accepted.
1. Authors need to focus on the abbreviations and use them in the first place.
2. For gold nanoparticles, term AuNPs has been used, however, at some points in the discussion, both terms have been used. Use AuNPs at all places.
3. In the beginning of section 3, a short paragraph on the synthesis and characterization of AuNPs would be a good option for readers.
4. Toxicity related concerns of AuNPs need to be captured.
5. Mechanism and challenges of sphere and rod shaped AuNPs in regards to the cellular targeting/internalization can be discussed in section 3.1/3.2.
6. Although there are several case studies captured in the review, it would be suitable to have one or two figure based reports to improve the quality and interest to the readers. For example, strong findings like tumor inhibition (AuNPs with Cisplatin) or presence of Au in tumors when NPs were conjugated with siRNA (Kumthekar et al.) or suppression of mRNA and protein expression level (AnuNPs coated with aminolevulinic acid) can be supported with a figure.
7. Authors need to discuss the limitations of AuNPs with respect to the scale up, cost and stability.
8. There is a typo in table 2 last column heading (NCT). Check for other typos.
Author Response
Reviewer #1 Current review article on the insights of gold NPs as a theranostic tool covers vast, latest, and quality compilation. Authors have presented a well drafted review by collecting information from highly scientific and reputed literature, while their method of selection was greatly unique.
Authors have widely focused on the different aspects of the gold NPs in terms of therapeutic approach, applications, current status and several case studies. Overall, the efforts in the current review are enormous.
R: Thank you very much. Our intention was to provide an outstanding information to the readers regarding this issue.
There are only few minor comments and suggestions to improve the final quality of the review after which it can be accepted.
R: Thank you.
1. Authors need to focus on the abbreviations and use them in the first place.
R: Thank you. They were corrected in the manuscript.
2. For gold nanoparticles, term AuNPs has been used, however, at some points in the discussion, both terms have been used. Use AuNPs at all places.
R: Thank you. They were replaced in the manuscript.
3. In the beginning of section 3, a short paragraph on the synthesis and characterization of AuNPs would be a good option for readers.
R: Thank you for this suggestion. They were introduced in the manuscript at lines 88 to 92, as seen below. AuNPs are metallic nanoparticles produced often through wet synthesis route. The shape and size of each kind of AuNPs depend on specific surfactants such as Sodium citrate and Cetyltrimethylammonium bromide (CTAB) which promotes anisotropic growth. There are some basic characterization procedures, based on spectroscopic techniques and electron microscopy imaging. Anatomical head and neck features as well as AuNPs synthesis and characterization are described below
4. Toxicity related concerns of AuNPs need to be captured.
R: Indeed. This issue has attracted the scientific and governmental attention to find out some suitable universal rules regarding the safety of AuNPs in medical practice. We added this information in the manuscript at lines 338-350, as highlighted bellow. Some level of cytotoxicity appears when AuNPs are used in biological assays. One common concern addresses tissue AuNPs aggregation that could lead to a local necrosis. Then, suitable chemical surface modifications with PEG has been applied to improve AuNPs bioavailability[65]. Additionally, AuNPs toxicity is also strongly related to by-products generated during synthesis and functionalization process, type of surfactants and their concentration, route of administration, their size, charge and shape[66]. For instance, it was already demonstrated that the smallest nanoparticles are the most cytotoxic[67,68]. Another important point is the lack of standard protocols for AuNPs synthesis and functionalization. Likewise, a well-established universal safety regulation based on reliable and robust multicentric research should be taken into account and, the international community must make an effort to adopt specific protocols for the characterization of these products[69]. In this way, the European approach to the regulatory testing of nanomaterials has tried to promote a wide consensus framework for the safety assessment of nanomaterials (NaNoREG)[70].
5. Mechanism and challenges of sphere and rod shaped AuNPs in regards to the cellular targeting/internalization can be discussed in section 3.1/3.2.
R: Thank you for your suggestion. They were added to this section at lines 130 to 141, as seen below: There are some natural mechanisms of AuNPs cellular targeting and internalization through cell membrane. The most common cellular internalization encompasses endocytic pathways such as clathrin and caveolin-dependent/-independent endocytosis, macropinocytosis and phagocytosis[20]. Besides, transmembrane receptors work to deliver and transport functionalized AuNPs increasing their cellular uptake[21]. Notwithstanding, several challenges exist for successful AuNPs cellular uptake related to their physicochemical properties, surface chemistry, stiffness, size and shapes. Additionally, human body barriers present remarkable biological features that may impar AuNPs internalization and trafficking. For instance, sphere-shaped AuNPs size and surface charge interactions can modify the desirable results of the different ionic interactions with cell membrane. For rod-shaped AuNPs to get a favorable uptake due to low surface energy requirements in cell membrane wrapping depends upon homogeneity and well controlled aspect-ratio synthesis, which have demonstrated to be challenging[19,20,22].
6. Although there are several case studies captured in the review, it would be suitable to have one or two figure based reports to improve the quality and interest to the readers. For example, strong findings like tumor inhibition (AuNPs with Cisplatin) or presence of Au in tumors when NPs were conjugated with siRNA (Kumthekar et al.) or suppression of mRNA and protein expression level (AnuNPs coated with aminolevulinic acid) can be supported with a figure.
R: It is a very good suggestion. Thank you. We provided 2 more figures illustrating the most important in vitro results applying AuNPs, pointed as Figure 3 and another one illustrating the most important in vivo data pointed as Figure 6. Both were described in the manuscript at lines 314-316 and 386-389, respectively.
7. Authors need to discuss the limitations of AuNPs with respect to the scale up, cost and stability.
R: This is an important issue and we have discussed it in the manuscript at lines 419-416. Thank you for suggest it. Another bottleneck is translating AuNPs from the laboratory scale to industry and up to now, there are only a few companies producing AuNPs at industrial level. Drawbacks such as batch to batch reproducibility is challenging[79]. The absence of standard operating procedures for synthesis, functionalization and characterization of AuNPs for biomedical applications, shortage and supply costs and colloidal stability must be largely discussed in an attempt to overcome them. Colloidal stability can be achieved since some parameters such as water quality, storage temperature and solution sterility are observed. Regarding shelf-time the long-term stability of sphere-shaped AuNPs coated with the monoclonal antibody Cetuximab was already observed up to 24 months[57].
8. There is a typo in table 2 last column heading (NCT). Check for other typos.
R: Thank you. It has already been corrected.
Reviewer 2 Report
The author presented An overview on using gold nanoparticles in treatment and diagnosis for neck and head cancers. The presented work is good in its presented form, but need more literature survey and the authors have to give their opinion about such topic.
Accept with minor revision...
Author Response
Reviewer #2
The author presented An overview on using gold nanoparticles in the treatment and diagnosis for neck and head cancers. The presented work is good in its presented form, but need more literature survey
R: In this state-of-the-art review we brought up the latest research regarding the feasibility of AuNPs to treat head and neck cancer. However, it is a relatively new issue and according to our searching parameters we have made a good literature review and the most relevant papers were cited and discussed. Nevertheless, we added new papers to support our discussion.
and the authors have to give their opinion about such topic.
R: Thank you. Indeed, we have not noticed this topic and now it has been added to the manuscript at lines 425-435 as highlighted below:
For the last five years, the amount of relevant published findings involving AuNPs and upper aerodigestive tract tumors has not been increased as they should, taking into account the advances in the technology available. Many data came from in vitro research that presents some limitations, for example, in terms of AuNPs biodistribution assessment. On the other hand, in vivo data still need different animal models focusing on long-term effects to support new clinical trials. Perhaps, the COVID-19 pandemic could be impacted researchers to continue work at their laboratories which may explain the reduction of publications in this issue. Nevertheless, it is still possible to assume that this field has a promising outcome. Although pre-clinical studies have suggested auspicious results and early-Phase I trials are currently ongoing, the translation to the clinic will demand more bio–nano interface research, circumventing possible systemic toxicities[77,78], and more clinical trials showing long-term follow-ups are required.
Accept with minor revision...
R: Thank you. The revision has been made.